# Clickbait vs. Quality: How Engagement-Based Optimization Shapes the Content Landscape in Online Platforms

*

## Abstract

Online content platforms commonly use engagement-based optimization when making recommendations. This encourages content creators to invest in quality, but also rewards gaming tricks such as clickbait. To understand the total impact on the content landscape, we study a game between content creators competing on the basis of engagement metrics and analyze the equilibrium decisions about investment in quality and gaming. First, we show the content created at equilibrium exhibits a *positive correlation between quality and gaming*, and we empirically validate this finding on a Twitter dataset. Using the equilibrium structure of the content landscape, we then examine the downstream performance of engagement-based optimization along two axes. Perhaps counterintuitively, the average quality of content consumed by users can decrease at equilibrium as gaming tricks become more costly for content creators to employ. Moreover, engagement-based optimization can perform worse in terms of user utility than a baseline with random recommendations. Altogether, our results highlight the need to consider content creator incentives when evaluating a platform's choice of optimization metric.

## CCS Concepts

• **Information systems** → **Content ranking**; • **Theory of computation** → **Market equilibria**.

## Keywords

content creator incentives, equilibrium characterization, societal impacts of online platforms

**ACM Reference Format:**
. 2018. Clickbait vs. Quality: How Engagement-Based Optimization Shapes the Content Landscape in Online Platforms. In *Proceedings of ACM Conference (Conference'17)*. ACM, New York, NY, USA, 17 pages. https://doi.org/XXXXXXX.XXXXXXX

## 1 Introduction

Content recommendation platforms typically optimize *engagement metrics* such as watch time, clicks, retweets, and comments (e.g., [40, 42]). Since engagement metrics increase with content quality, one might hope that engagement-based optimization would lead to desirable recommendations. However, engagement-based optimization has led to a proliferation of clickbait [29], incendiary content [33], divisive content [37] and addictive content [9]. A driver of

these negative outcomes is that engagement metrics not only reward *quality*, but also reward *gaming tricks* such as clickbait that worsen the user experience.

In this work, we examine how engagement-based optimization shapes the landscape of content available on the platform. We focus on the role of strategic behavior by content creators: competition to appear in a platform's recommendations influences what content they are incentivized to create [6, 20, 23]. In the case of engagement-based optimization, we expect that creators strategically decide how much effort to invest in quality versus how much effort to spend on gaming tricks, both of which increase engagement. For example, since the engagement metric for Twitter includes the number of retweets [42]—which includes both quote retweets (where the retweeter adds a comment) and non-quote retweets (without any comment)—creators can either increase quote retweets by using offensive or sensationalized language [32] or increase non-quote retweets by putting more effort into the quality of their content (Example 1). When the engagement metric for video content includes total watch time [40], creators may either increase the "span" of their videos—by investing in quality—or instead increase the "moreishness" by leveraging behavioral weaknesses of users such as temptation [25] (Example 2). When the engagement metric includes clicks, creators can rely on clickbait headlines [29] or actually improve content quality (Example 3).

Intuitively, creators must balance two opposing forces when incorporating quality and gaming tricks in the content that they create. On one hand, it is expensive for creators to invest in quality, but it may be much cheaper to utilize gaming tricks that also increase engagement. On the other hand, gaming tricks generate disutility for users, which might discourage them from engaging with the content even if it is recommended by the platform. This raises the questions: *Under engagement-based optimization, how do creators balance between quality and gaming tricks at equilibrium? What is the impact on the content landscape and on the downstream performance of engagement-based optimization?*

To investigate these questions, we propose and analyze a game between content creators competing for user consumption through a platform that performs engagement-based optimization. We model the content creator as jointly choosing investment in quality and utilization of gaming tricks. Both quality and gaming tricks increase engagement from consumption, and utilizing gaming tricks is relatively cheaper for the creators than investing in quality. However, gaming decreases user utility, while quality increases user utility, and a user will not consume the content if their utility from consumption is negative. We study the Nash equilibrium in the game between the content creators.

We first examine the balance between gaming tricks and quality amongst content created at equilibrium (Section 3). Interestingly, we find that there is a *positive correlation* between gaming and investment at equilibrium: higher-quality content typically exhibits *higher* levels of gaming tricks. We prove that equilibria exhibit this

positive correlation (Figure 1a; Theorem 2), and we also empirically validate this finding on a Twitter dataset [31] (Figure 2 and Table 1). These results suggest that gaming tricks and quality should be viewed as *complements*, rather than substitutes.

Accounting for how the platform's metric shapes the content landscape at equilibrium, we then analyze the downstream performance of engagement-based optimization (Section 4). We uncover striking properties of engagement-based optimization along two performance axes and discuss implications for platform design.

- **Content Quality.** First, we examine the average quality of content consumed by users and show that it can *decrease* as gaming tricks become more costly for creators (Figure 1b; Theorem 3). In other words, as it becomes more difficult for content creators to game the engagement metric, the average content quality at equilibrium becomes worse. From a platform design perspective, this suggests that increasing the transparency of the platform's metric (which intuitively reduces gaming costs for creators) may improve the average quality of content consumed by users.
- **User Welfare.** We next examine the user welfare at equilibrium. We show that engagement-based optimization can lead to lower user welfare at equilibrium than even the conservative baseline of randomly recommending content (Figure 1c; Theorem 5). From a platform design perspective, this suggests that engagement-based optimization may not retain users in a competitive marketplace in the long-run.

Altogether, these results illustrate the importance of factoring in the endogeneity of the content landscape when assessing the downstream impacts of engagement-based optimization.

## 1.1 Related Work

Our work connects to research threads on *content creator competition in recommender systems* and *strategic behavior in machine learning*.

*Content-creator competition in recommender systems.* An emerging line of work has proposed game-theoretic models of content creator competition in recommender systems, where content creators strategically choosing what content to create [4, 6, 8] or the quality of their content [15, 36]. Some models embed content in a continuous, multi-dimensional action space, characterizing when specialization occurs [23] and the impact of noisy recommendations [20]. Other models capture that content creators compete for engagement [43] and general functions of platform "scores" across the content landscape [44]. These models have also been extended to dynamic settings, including where the platform learns over time [14, 21, 28] and where content providers learn over time [8, 35]. However, while these works all assume that creator utility depends only on winning recommendations (or only on content scores according to the platform metric [43, 44]), our model incorporates *misalignment* between the platform's (engagement) metric and user utility.[1] In particular, our model and insights rely on the fact that creators only derive utility if their content is recommended *and* the content generates nonnegative user utility.

---

[1] A rich line of work (e.g., [11, 25, 30, 41]) has identified sources of misalignment between engagement metrics and user utility and broader issues with inferring user preferences from observed behaviors; these sources of misalignment motivated us to incorporate gaming tricks which increase engagement but reduce user utility into our model.

Several other works study content creator competition under different modelling assumptions: e.g., where content quality is fixed and all creator actions are gaming [32], where content creators have fixed content but may dynamically leave the platform over time [7], where the platform designs a contract determining payments and recommendations [45], where the platform creating its own content [3], and where the platform designs badges to incentivize user-generated content [22]. This line of work also builds on Hotelling models of product selection from economics (e.g. [19, 38], see Anderson et al. [2] for a textbook treatment).

*Strategic behavior in machine learning.* A rich line of work on *strategic classification* (e.g. [10, 17]) focuses primarily agents strategically adapting their features in classification problems, whereas our work focuses on agents competing to win users in recommender systems. Some works also consider improvement (e.g. [1, 16, 24]), though also with a focus on classification problems. One exception is [27], which studies ranking problems; however, the model in [27] considers all effort as improvement, whereas our model distinguishes between clickbait and quality. Other topics studied in this research thread include shifts to the population in response to a machine learning predictor (e.g. [34]), strategic behavior from users (e.g. [18]), and incentivizing exploration ([12, 26, 39]).

## 2 Model

We study a stylized model for content recommendation in which an online platform recommends to a single user a single piece of digital content within the content landscape available on the platform.[2] There are $P \geq 2$ content creators who each create a single piece of content and compete to appear in recommendations. Building on the models of Ben-Porat and Tennenholtz [6], Hron et al. [20], Jagadeesan et al. [23], Yao et al. [44], the content landscape is *endogenously* determined by the multi-dimensional actions of the content creators.

### 2.1 Creator Costs, User Utility, and Platform Engagement

Since our focus is on investment versus gaming, we project pieces of digital content into 2 dimensions $w = [w_{\text{costly}}, w_{\text{cheap}}] \in \mathbb{R}^2_{\geq 0}$. The more costly dimension $w_{\text{costly}}$ denotes a measure of the content's quality, whereas the cheap dimension $w_{\text{cheap}}$ reflects the extent of gaming tricks present in the content. These measures are normalized so that $w = [0, 0]$ represents content generated by a creator who exerted no effort on quality or gaming. The costly and cheap dimensions impact *creator costs*, *user utility*, and *platform engagement* in different ways, as we specify below.

*Creator Costs.* Each content creator pays a (one-time) cost of $c(w) \geq 0$ to create content $w \in \mathbb{R}^2_{\geq 0}$. We assume that $c$ is continuously differentiable in $w$ and satisfies the following additional assumptions. First, investing in quality content is costly: $(\nabla(c(w)))_1 > 0$ for all $w \in \mathbb{R}^2_{\geq 0}$. Moreover, engaging in gaming tricks is either always free or always incurs a cost: either $(\nabla(c(w)))_2 > 0$ for all $w \in \mathbb{R}^2_{\geq 0}$ or $(\nabla(c(w)))_2 = 0$ for all $w \in \mathbb{R}^2_{\geq 0}$. Finally, producers

---

[2] Our model can also capture a stream of content and a population of homogeneous users, even though we abstract away from this by focusing on one recommendation to a single user at a time.

have the option to opt out by not investing costly effort in either gaming tricks or quality: $c([0,0]) = 0$.

*User Utility.* The user has relative tolerance for gaming tricks, specified by a type $t > 0$. The user receives utility $u(w, t) \in \mathbb{R}$ from consuming content $w \in \mathbb{R}_{\geq 0}^2$, where the utility function is normalized so that the user's outside option offers 0 utility. We assume that $u$ is continuously differentiable in $w$ and satisfies the following additional assumptions. The user derive positive utility from $w_{\text{costly}}$ and negative utility from $w_{\text{cheap}}$:

- For $w_{\text{costly}} \in \mathbb{R}_{\geq 0}$: the utility $u([w_{\text{costly}}, w_{\text{cheap}}], t)$ is strictly decreasing in $w_{\text{cheap}}$ and approaches $-\infty$ as $w_{\text{cheap}} \to \infty$.
- For $w_{\text{cheap}} \in \mathbb{R}_{\geq 0}$: the utility $u([w_{\text{costly}}, w_{\text{cheap}}], t)$ is strictly increasing in $w_{\text{costly}}$ and approaches $\infty$ as $w_{\text{costly}} \to \infty$.

*Engagement.* If the user chooses to consume content $w$, this interaction generates platform engagement $M^{\text{E}}(w) \in \mathbb{R}$. The engagement metric $M^{\text{E}}(w)$ depends on the content $w$ but is independent of the user's type $t$ (conditional on the user choosing to consume the content). We assume that $M^{\text{E}}$ is continuously differentiable in $w$ and satisfies the following additional assumptions. First, both cheap gaming tricks and investment in quality increase the engagement metric: $(\nabla M^{\text{E}}(w))_1, (\nabla M^{\text{E}}(w))_2 > 0$ for all $w \in \mathbb{R}_{\geq 0}^2$. Moreover, the engagement metric is nonnegative: $M^{\text{E}}(w) \geq 0$ for all $w \in \mathbb{R}_{\geq 0}^2$. Finally, the relative cost of gaming tricks versus costly investment is less than the relative benefit: $\frac{(\nabla c(w))_2}{(\nabla c(w))_1} < \frac{(\nabla M^{\text{E}}(w))_2}{(\nabla M^{\text{E}}(w))_1}$ for all $w \in \mathbb{R}_{\geq 0}^2$. In other words, it is more cost-effective for a creator to increase the engagement metric via gaming than via quality, for a user who would choose to consume the content either way.

## 2.2 Timing and Interaction between the Platform, Users, and Content Creators

The interaction between the platform, users, and content creators defines a game that proceeds in stages. The timing is as follows:

**Stage 1:** Each content creator $i \in [P]$ simultaneously chooses what content $w_i \in \mathbb{R}_{\geq 0}^2$ to create. These choices give rise to a content landscape $\mathbf{w} = (w_1, \ldots, w_P)$.

**Stage 2:** A user with type $t$ comes to the platform.

**Stage 3:** The platform observes the user's type $t$ and evaluates content $w$ according to a metric $M : \mathbb{R}_{\geq 0}^2 \to \mathbb{R}$ that maps each piece of content $w_i$ to a score $M(w_i)$. The platform optimizes $M$ over content available in the content landscape that generates nonnegative utility for the user. More formally, the platform selects content creator

$$i^*(M; \mathbf{w}) \in \underset{i \in [P]}{\arg\max}(M(w_i) \cdot \mathbb{1}[u(w_i, t) \geq 0]),$$

breaking ties uniformly at random, and recommends the content $w_{i^*(M; \mathbf{w})}$ to the user.

**Stage 4:** The user consumes the the recommended content $w_{i^*(M; \mathbf{w})}$ if and only if $u(w_{i^*(M; \mathbf{w})}, t) \geq 0$ (i.e., if and only if the content is at least as appealing as their outside option).

We assume that content creators, the platform, and the user all know the type $t$ and the user utility function $u(\cdot, t)$. Moreover, the platform can observe observe the full content landscape $\mathbf{w}$. This provides the platform with sufficient information to solve the optimization problem $\arg\max_{i \in [P]}(M(w_i) \cdot \mathbb{1}[u(w_i, t) \geq 0])$ in

**Stage 3.**[3] The user can also observe the content $w$ recommended to them, so they can evaluate whether $u(w_{i^*(M; \mathbf{w})}, t) \geq 0$.[4]

*Equilibrium decisions of content creators.* The recommendation process defines a game played between the content providers, who strategically choose their content $w_i \in \mathbb{R}_{\geq 0}^2$ in **Stage 1**. We assume that values are normalized so that a content creator receives a value of 1 for being shown to a user. Since the goods are digital, production costs are one-time and incurred regardless of whether the user consumes the content. Producer $i$'s expected utility is therefore

$$U_i(w_i; \mathbf{w}_{-i}) := \mathbb{E}[\mathbb{1}[i^*(M; \mathbf{w}) = i]] - c(w), \tag{1}$$

where the expectation is over the randomness of tiebreaking by the platform. We allow content creators to randomize over their choice of content, and write $\mu_i \in \Delta(\mathbb{R}_{\geq 0}^2)$ for such a mixed strategy. A (mixed) Nash equilibrium $(\mu_1, \ldots, \mu_P)$, for $\mu_i \in \Delta(\mathbb{R}_{\geq 0}^2)$, is a profile of mixed strategies that are mutual best-responses. Since the content creators are symmetric in our model, we will focus primarily on symmetric mixed Nash equilibria in which each creator employs the same mixed strategy, which must exist (see Theorem 1 below). Note that the Nash equilibrium specifies the distribution over content landscapes $\mathbf{w}$.

*The platform's choice of metric $M$ in Stage 3.* We primarily focus on *engagement-based optimization* where $M = M^{\text{E}}$, meaning that the platform optimizes for engagement. As a benchmark, we also consider *investment-based optimization* where $M(w) = M^{\text{I}}(w) := w_{\text{costly}}$ does not reward gaming tricks; however, note that this baseline is idealized, since $w_{\text{costly}}$ is not always identifiable from observable data in practice. As another baseline, we consider *random recommendations* where $M(w) = M^{\text{R}}(w) := 1$ which captures choosing uniformly at random from all content that generates nonnegative user utility.

## 2.3 Running examples

We provide instantiations of our models that serve as running examples throughout the paper.

**Example 1.** *Consider an online platform such as Twitter which uses retweets as one of the terms its objective [42]. However, Twitter does not differentiate between quote retweets (where the retweeter adds a comment) and non-quote retweets (where there is no added comment). Creators can cheaply increase quote retweets by increasing the offensiveness or sensationalism of the content [31], or increase non-quote retweets by actually improving content quality. As a stylized model for this, let $w_{cheap}$ be the offensiveness of the content and let $w_{costly}$ capture costly investment into content quality. Let the utility function of a user with type $t > 0$ be the linear function $u(w, t) = w_{costly} - (w_{cheap}/t) + \alpha$, where $\alpha \in \mathbb{R}$ is the baseline utility from no effort and $t$ captures the user's tolerance to offensive content.*

---

[3]The platform may be able to evaluate $\arg\max_{i \in [P]}(M(w_i) \cdot \mathbb{1}[u(w_i, t) \geq 0])$ with less information. For example, if $M = M^{\text{E}}$, then $M^{\text{E}}(w)$ can typically be estimated from observable data such as user behavior patterns without knowledge of $w_{\text{costly}}$ and $w_{\text{cheap}}$. Moreover, since $\mathbb{1}[u(w_i, t) \geq 0]$ captures the event that users click on the content $w_i$, if the platform has a predictor for clicks, this would provide them an estimate of $\mathbb{1}[u(w_i, t) \geq 0]$.

[4]In reality, the user may not always be able to perfectly observe $w_{\text{costly}}$ and $w_{\text{cheap}}$ (or gauge their own utility) without consuming the content. Our model makes the simplifying assumption that user choice is noiseless.

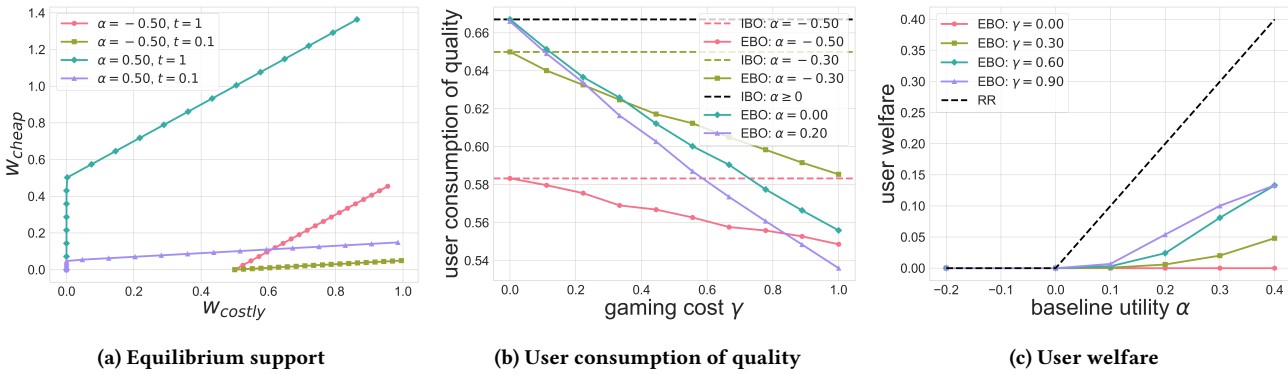

(a) Equilibrium support           (b) User consumption of quality           (c) User welfare

**Figure 1: Analysis of symmetric mixed equilibria for engagement-based optimization (EBO) in Example 1.** *Left:* **Equilibrium support for $\gamma = 0.1$. The support exhibits positive correlation between gaming tricks $w_{\text{cheap}}$ and investment in quality $w_{\text{costly}}$ (Theorem 2). The slope varies with the type $t$ and the intercept varies with the baseline utility $\alpha$ (Theorem 1).** *Middle and right:* **Equilibrium performance for $P = 2$ producers along user consumption of quality (middle) and user welfare (right). The performance is numerically estimated from 100,000 samples from the equilibrium distributions (Section 2.4). The parameter settings are $t = 1$ (left) and $t = 5$ (right). The equilibrium performance of investment-based optimization (IBO) and random recommendations (RR) are analytically computed from the equilibrium distributions (Section 4.3) and shown as baselines. User consumption of quality can decrease with gaming costs (left; Theorems 3-4), and user welfare can be lower for EBO than for RR (right; Theorem 5).**

Let the platform metric $M^E(w) = w_{\text{costly}} + w_{\text{cheap}}$ and cost function $c(w) = w_{\text{costly}} + \gamma \cdot w_{\text{cheap}}$ for $\gamma \in [0, 1)$ also be linear functions. The platform metric captures the idea that the platform does not distinguish between different types of retweets; the cost function captures the idea that it is relatively easier for producers to insert sensationalism into tweets, which requires just a few word changes, compared to improving content quality, which might require, for example, time-intensive fact-checking.

**Example 2.** *Consider an online platform such as TikTok [40] that incorporates watch time into its objective. Creators can increase watch time by: a) creating "moreish" content that keeps people watching a video even after they are deriving disutility from it, or b) increasing "span" by increasing the amount of substantive content, as modelled in Kleinberg et al. [25]. More formally, let $w_{\text{costly}} := \frac{q}{1-q}$ be a reparameterized version of the span $p \in [0, 1]$, let $w_{\text{cheap}} := \frac{p}{1-p}$ be a reparameterized version of the moreishness $q \in [0, 1]$. For a given user, let $v$ be the value derived from each time step from watching substantive content, let $W$ be the outside option for each time step, and let $t := v/W - 1 > 0$ capture the shifted ratio. In this notation, the engagement metric $M^E$ and user utility $u$ from Kleinberg et al. [25] take the following form: $M^E([w_{\text{cheap}}, w_{\text{costly}}]) := w_{\text{costly}} + w_{\text{cheap}} + 1$ and $u(w, t) := W \cdot t \cdot \left(w_{\text{costly}} - w_{\text{cheap}}/t + 1\right)$ We further specify the cost function based on a linear combination of the expected amount of "span" time and the expected amount of "moreish" time that the user consumes: $c(w) := w_{\text{costly}} + \gamma \cdot w_{\text{cheap}}$ where $\gamma \in [0, 1)$ specifying the cost of increasing moreishness relative to increasing span.*

**Example 3.** *Consider an online platform such as YouTube that historically used clicks as one of the terms in their objective. Creators can cheaply increase clicks by leveraging clickbait titles or thumbnails [29] or by increasing the quality of their content. As a stylized model*

for this, let $w_{\text{cheap}}$ capture how flashy or sensationalized the title or thumbnail is, and let $w_{\text{costly}}$ capture the quality of the content.

## 2.4 Equilibrium characterization

We show by construction that a symmetric mixed equilibrium exists for engagement-based optimization for arbitrary setups. The game has a complex structure due to its infinite action space and discontinuous utility functions. Nonetheless, for each possible setting of $P$, $c$, $u$, and $t$, we construct a symmetric mixed equilibria with a clean closed-form characterization (Figure 1a).

To state our characterization, we define a distribution $\mu^e(P, c, u, t)$ over $\mathbb{R}^2_{\geq 0}$. To simplify the notation, we convert the two-dimensional action space into a one-dimensional curve that specifies the support of the equilibrium (Figure 1a). We define the *minimum-investment function* $f_t : \mathbb{R}_{\geq 0} \to \mathbb{R}_{\geq 0}$, which captures the amount of investment in quality needed to offset the disutility from gaming tricks, as follows:

$$f_t(w_{\text{cheap}}) := \inf \left\{ w_{\text{costly}} \mid w_{\text{costly}} \geq 0, u([w_{\text{costly}}, w_{\text{cheap}}], t) \geq 0 \right\}. \tag{2}$$

Within the one-dimensional curve, the content $w$ is entirely specified by the cheap component $w_{\text{cheap}}$, which motivates us to define a one-dimensional cost function for content along each curve:

$$C_t(w_{\text{cheap}}) := c([f_t(w_{\text{cheap}}), w_{\text{cheap}}]). \tag{3}$$

For example, the functions $f_t$ and $C_t$ take the following form in Example 1:

**Example 1** (Continued). *The functions $f_t$ and $C_t$ are as follows:*

$$f_t(w_{\text{cheap}}) = \max(0, (w_{\text{cheap}}/t) - \alpha)$$

$$C_t(w_{\text{cheap}}) = \begin{cases} w_{\text{cheap}}(\gamma + 1/t) - \alpha & \text{if } w_{\text{cheap}} > \max(0, t \cdot \alpha) \\ w_{\text{cheap}} \cdot \gamma & \text{if } w_{\text{cheap}} \leq t \cdot \alpha. \end{cases}$$

*As $t$ increases (and users becomes more tolerant to gaming tricks), the slope of $f_t$ and $C_t$ both decrease. The minimum-investment $f_t$ is independent of $\gamma$, but the cost function increases with $\gamma$.*

We define $(W_{\text{cheap}}, W_{\text{costly}}) \sim \mu^{\text{e}}(P, c, u, t)$ as follows. The marginal distribution $W_{\text{cheap}}$ is

$$\mathbb{P}[W_{\text{cheap}} \leq w_{\text{cheap}}] = \left(\min(1, C_t(w_{\text{cheap}}))\right)^{1/(P-1)}.$$

For each $w_{\text{cheap}} \in \text{supp}(W_{\text{cheap}})$, the conditional distribution $W_{\text{costly}} \mid W_{\text{cheap}} = w_{\text{cheap}}$ is a point mass at $f_t(w_{\text{cheap}})$ if $w_{\text{cheap}} > 0$ and a point mass at 0 if $w_{\text{cheap}} = 0$. For example, the distribution takes the following form within Example 1.

**Example 1** (Continued). *Let $P = 2$, $\gamma = 0.1$, and $\alpha = 0.5$. Then, $W_{\text{cheap}}$ and $W_{\text{costly}}$ are both distributed as uniform distributions and $\mu^e(P, c, u, t)$ is supported on a line segment (Figure 1a).*

We prove that $\mu^{\text{e}}(P, c, u, t)$ is a symmetric mixed equilibrium.[5]

**Theorem 1.** *The distribution $\mu^e(P, c, u, t)$ is a symmetric mixed equilibrium in the game with $M = M^E$.*

The proof is deferred to Appendix B.

We also compute a symmetric mixed equilibrium for investment-based optimization and random recommendations (Section 4.3).

## 3 Positive correlation between quality and gaming tricks

When the platform optimizes engagement metrics $M^{\text{E}}$, each content creator *jointly* determines how much to utilize gaming tricks and invest in quality. The creators' equilibrium decisions of how to balance gaming and quality in turn determine the properties of content in the content landscape. In this section, we show that there is a positive correlation between gaming and quality: that is, content that exhibits higher levels of gaming typically exhibits higher investment in quality. We prove that the equilibria satisfy this property (Section 3.1), and we empirically validate this property on a dataset [31] of Twitter recommendations (Section 3.2).

### 3.1 Theoretical analysis of balance between gaming and quality

We theoretically analyze the balance of gaming and quality at equilibrium as follows. Since the content landscape $\mathbf{w} = [w_1, \ldots, w_P]$ at equilibrium consists of content $w_i \sim \mu_i$ for $i \in [P]$, the set of content that shows up in the content landscape with nonzero probability is equal to $\cup_{i \in [P]} \text{supp}(\mu_i)$. We examine the relationship between the quality $w_{\text{costly}}$ and the level of gaming $w_{\text{cheap}}$ for $w \in \cup_{i \in [P]} \text{supp}(\mu_i)$.

We show that for any (possibly asymmetric) mixed strategy equilibrium[5], the support satisfies the following *positive correlation* property: a creator's investment in quality content weakly increases with the creator's utilization of gaming tricks (Figure 1a).

**Theorem 2.** *Suppose that gaming is not costless (i.e. $(\nabla(c(w)))_2 > 0$ for all $w \in \mathbb{R}_{\geq 0}^2$). Let $(\mu_1, \mu_2, \ldots, \mu_P)$ be any mixed Nash equilibrium, and let $w^1, w^2 \in \cup_{i \in [P]} \text{supp}(\mu_i)$ be any two pieces of content in the support. If $w_{\text{cheap}}^2 \geq w_{\text{cheap}}^1$, then $w_{\text{costly}}^2 \geq w_{\text{costly}}^1$.*

Theorem 2 illustrates a positive correlation between gaming tricks and investment in quality in the content landscape. Perhaps surprisingly, this positive correlation indicates that even high-quality content on the content landscape will have clickbait headlines or exhibit other gaming tricks. Thus, gaming tricks and investment should be viewed as *complements* rather than substitutes.

We provide a proof sketch of Theorem 2.

PROOF SKETCH OF THEOREM 2. For the symmetric mixed equilibrium $\mu^{\text{e}}(P, c, u, t)$, this result can derived from the closed-form characterization along with the fact that $f_t$ is weakly increasing in $w_{\text{cheap}}$. We generalize this to any possibly asymmetric mixed equilibrium by showing that the support $\cup_{i \in [P]} \text{supp}(\mu_i)$ is always contained in:

$$\underbrace{\{(f_t(w_{\text{cheap}}), w_{\text{cheap}}) \mid w_{\text{cheap}} \geq 0\}}_{(A)} \cup \underbrace{\{(0, 0)\}}_{(B)},$$

where $f_t$ is the minimum-investment function. The set (A) corresponds to investing the minimum amount in quality to maintain nonnegative user utility, and The set (B) captures creators "opting out" of the game by not expending any costly effort in producing their content. The intuition for $\cup_{i \in [P]} \text{supp}(\mu_i)$ being contained in the union of (B) and (A) is that a creator is either incentivized to opt out or to set $w_{\text{costly}}$ as low as possible to achieve nonnegative user utility. We defer the full proof to Appendix C. □

### 3.2 Empirical analysis on Twitter dataset

We next provide empirical validation for the positive correlation between gaming and investment on a Twitter dataset [31]. The dataset consists of survey responses from 1730 participants, each of whom was asked several questions about each of the top ten tweets in their personalized and chronological feeds. Using the user survey responses, we associate each tweet with a tuple:

$$(f, g, a, l) \in \{E, C\} \times \{P, \neg P\} \times \{0, 1, 2, 3, 4\} \times \mathbb{Z}_{\geq 0}.$$

The *feed* $f \in \{E, C\}$ captures whether the tweet was in the user's engagement-based feed ($f = E$) or chronological feed ($f = C$). The *genre* $g \in \{P, \neg P\}$ captures whether the user labelled the content as in the political genre ($g = P$) or not ($g = \neg P$). The *angriness level* $a \in \{0, 1, 2, 3, 4\}$ captures the reader's evaluation of how angry the author appears in their tweet, rated numerically between 0 and 4.[6] The *number of favorites* $l \in \mathbb{Z}_{\geq 0}$ captures the number of favorites (i.e. "heart reactions") of the tweet. Let $D$ be the multiset $D$ of tuples from the tweets in the dataset, and let $\mathcal{D}$ be the distribution where $(f, g, a, l)$ is drawn uniformly from the multiset $D$.

We map this empirical setup to Example 1 as follows. Since $w_{\text{cheap}}$ is intended to capture the offensiveness of content in Example 1, we estimate $w_{\text{cheap}}$ by the angriness level $a$. Since $w_{\text{costly}}$ is intended to capture the costly investment into content quality in Example 1, we estimate $w_{\text{costly}}$ by the number of favorites $l$. We expect that increasing author angriness $w_{\text{cheap}}$ decreases user utility, drawing upon intuition from Munn [33] that incendiary or divisive content drives engagement by provoking outrage in users. Furthermore, we expect that higher quality content would generally receive more favorites $w_{\text{costly}}$ and lead to higher user utility.

---

[5]There can exist asymmetric equilibria: for example, if $P = 3$, the mixed strategy profile where $\mu_1$ is a point mass at $[0, 0]$ and $\mu_2 = \mu_3 = \mu^{\text{e}}(2, c, u, t)$ is an equilibrium.

[6]The survey question asked: "How is [author-handle] feeling in their tweet?" [31]

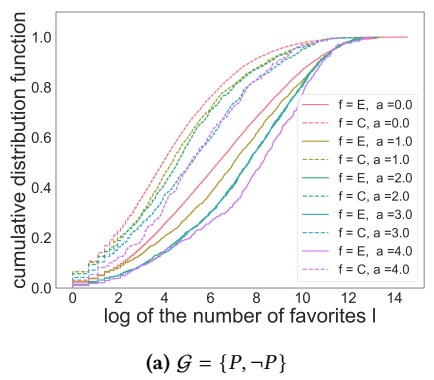

**(a)** $\mathcal{G} = \{P, \neg P\}$

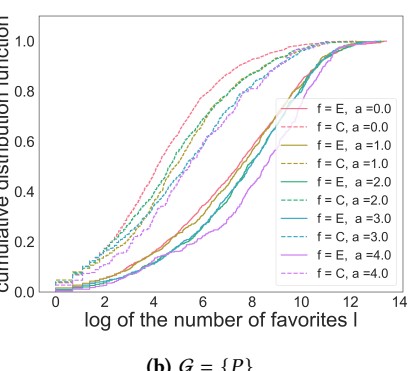

**(b)** $\mathcal{G} = \{P\}$

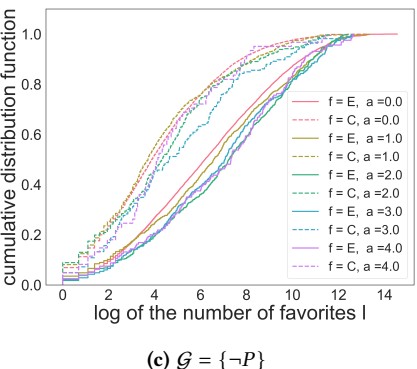

**(c)** $\mathcal{G} = \{\neg P\}$

**Figure 2: Cumulative distribution function $H_{a,f,\mathcal{G}}$ of the number of favorites ($w_{\text{costly}} = l$) conditioned on different angriness levels ($w_{\text{cheap}} = a$) on a dataset [31] of tweets from the engagement-based feeds ($f = E$) and chronological feeds ($f = C$). The tweet genre is unrestricted (left), restricted to political tweets (middle), and restricted to not political tweets (right). The cdf for higher values of $a$ appears to stochastically dominate the cdf for lower values of $a$, suggesting a positive correlation between $w_{\text{cheap}}$ and $w_{\text{costly}}$. The stochastic dominance is more pronounced for political tweets than for non-political tweets, and it occurs for engagement-based and chronological feeds.**

|  | $\mathcal{G} = \{P, \neg P\}$ | $\mathcal{G} = \{P\}$ | $\mathcal{G} = \{\neg P\}$ |
|---|---|---|---|
| $f = E$ | 0.131 | 0.092 | 0.048 |
|  | $(2 \cdot 10^{-76})$ | $(2 \cdot 10^{-10})$ | $(3 \cdot 10^{-9})$ |
| $f = C$ | 0.086 | 0.138 | 0.004 |
|  | $(2 \cdot 10^{-33})$ | $(4.49 \cdot 10^{-19})$ | $(3 \cdot 10^{-1})$ |

**Table 1: Correlation coefficient $\rho_{f,\mathcal{G}}$ (with $p$-value $p_{f,\mathcal{G}}$ in parentheses) between the number of favorites ($w_{\text{costly}} = l$) and the angriness level ($w_{\text{cheap}} = a$) on a dataset [31] of tweets from the engagement-based feeds ($f = E$) and chronological feeds ($f = C$) and across political ($P$) and non-political ($\neg P$) tweets. The correlation coefficient is positive (though weak) and statistically significant in all cases except for non-political tweets in the chronological feed. Moreover, correlations are stronger for political than for non-political tweets.**

We analyze the relationship between the number of favorites ($w_{\text{costly}}$) and the angriness ($w_{\text{cheap}}$) with two different approaches:

- *Stochastic dominance of conditional distributions:* Given an angriness level $a \in \{0, 1, 2, 3, 4\}$, feed $f \in \{E, C\}$ and subset of genres $\mathcal{G} \subseteq \{P, \neg P\}$, consider the random variable $\ln(L)$ where $(F, G, A, L)$ is drawn from the conditional distribution $\mathcal{D} \mid (A = a, F = f, G \in \mathcal{G})$. We let $H_{a,f,\mathcal{G}}$ denote the cumulative density function of this random variable. We visually examine the extent to which $H_{a,f,\mathcal{G}}$ stochastically dominates $H_{a',f,\mathcal{G}}$ when $a > a'$.

- *Correlation coefficient:* Given a feed $f \in \{E, C\}$ and subset of genres $\mathcal{G} \subseteq \{P, \neg P\}$, we compute the multiset

$$S_{f,\mathcal{G}} := \left\{ (a, l) \mid (f, g, a, l) \in D \mid f = f', g' \in \mathcal{G} \right\} \quad (4)$$

We compute the Spearman's rank correlation coefficient $\rho_{f,\mathcal{G}} \in [-1, 1]$ of the multiset $S_{f,\mathcal{G}}$ and a corresponding p-value $p_{f,\mathcal{G}}$.[7]

*Stochastic dominance of conditional distributions.* Figure 2 shows the cumulative distribution function $H_{a,f,\mathcal{G}}$ for different values of $a$, $f$, and $\mathcal{G}$. The primary finding is that in all of the plots, the cdf for higher values of $a$ visually appears to stochastically dominate the cdf for lower values of $a$. This stochastic dominance reflects a higher author's angriness level $w_{\text{cheap}} = a$ leads to higher numbers of favorites $w_{\text{costly}} = l$, thus suggesting that content with higher levels of gaming $w_{\text{cheap}}$ also exhibit higher quality $w_{\text{costly}}$.

Interestingly, the stochastic dominance is most pronounced when $\mathcal{G} = \{P, \neg P\}$ and $\mathcal{G} = \{P\}$, but less pronounced when $\mathcal{G} = \{\neg P\}$. This aligns with the intuition that increasing author angriness more effectively increases engagement for political tweets than for non-political tweets.[8] Moreover, within $\mathcal{G} = \{P, \neg P\}$ and $\mathcal{G} = \{P\}$, the stochastic dominance occurs for both $f = E$ and $f = C$. We view each of $f = E$ and $f = C$ as capturing a different slice of the content landscape: the fact that stochastic dominance occurs in two different slices suggests it broadly occurs in the content landscape.

*Correlation coefficient.* Table 1 shows $\rho_{f,\mathcal{G}}$ for different genres of tweets and feeds. Interestingly, the correlation coefficient is positive in all cases, which suggests that content with higher levels of gaming tend to exhibit higher levels of investment in quality. However, the correlation is somewhat weak: this may be due to angriness ratings being incomparable across different survey participants. Nonetheless, the correlation is stronger for political content, which again aligns with the intuition that increasing author angriness is more effective in increasing engagement for political tweets.

## 4 Performance of engagement-based optimization at equilibrium

In this section, taking into account the structure of the the content landscape at equilibrium, we investigate the downstream performance of engagement-based optimization. As baselines, we consider investment-based optimization (an idealized baseline that

---

[7]The p-value is for a one-sided hypothesis test with null hypothesis that $a$ and $l$ have no ordinal correlation, calcuated using the scipy.stats.spearmanr Python library.

[8]For non-political tweets, we expect other types of gaming tricks are employed.

optimizes directly for quality $M^I(w) = w_{costly}$) and random recommendations (a trivial baseline that results in randomly choosing from content that achieves nonnegative user utility). We highlight striking aspects of these comparisons (Figures 1b-1c), considering two qualitatively different performance axes: user consumption of quality (Section 4.1) and user utility (Section 4.2).

Our comparisons take into account the *endogeneity of the content landscape*: i.e., that the content landscape at equilibrium depends on the choice of metric. The possibility of multiple equilibria casts ambiguity on which equilibrium to consider.[5] To resolve this ambiguity, we focus on the (symmetric mixed) equilibria in our characterization results throughout this section. For engagement-based optimization, we focus on $\mu^e(P, c, u, t)$ as defined in Section 2.4. For our baseline approaches, we characterize a symmetric mixed equilibrium $\mu^i(P, c, u, t)$ for investment-based optimization and a symmetric mixed equilibrium $\mu^r(P, c, u, t)$ for random recommendations in Section 4.3.

Our results in this section focus on Example 1 across different settings of the baseline utility $\alpha$ and gaming cost $\gamma$. Our results hold for any number of producers $P \geq 2$ and any type $t > 0$.

## 4.1 User consumption of quality

We first consider the average quality of content consumed by the user (formalized below), focusing on Example 1. We show that as gaming costs increase, the performance of engagement-based optimization *worsens*; in fact, unless gaming is *costless*, engagement-based optimization performs strictly worse than investment-based optimization.

We formalize user consumption of quality by:

$$\text{UCQ}(M; \mathbf{w}) := \mathbb{E}\left[M^I(w_{i^*(M;\mathbf{w})}) \cdot \mathbb{1}\left[u(w_{i^*(M;\mathbf{w})}, t) \geq 0\right]\right],$$

which only counts content quality if the content is actually consumed by the user. Taking into account the endogeneity of the content landscape, the user consumption of quality at a symmetric mixed Nash equilibrium $\mu^M$ is:

$$\mathbb{E}_{\mathbf{w} \sim (\mu^M)^P}\left[\text{UCQ}(M; \mathbf{w})\right].$$

The following result shows that in Example 1 the average user consumption of quality strictly *decreases* as gaming costs (parameterized by $\gamma$) become more expensive (Figure 1b).

**Theorem 3.** *For any sufficiently large baseline utility $\alpha > -1$ and for bounded gaming costs $\gamma \in [0, 1)$, the user consumption of quality $\mathbb{E}_{\mathbf{w} \sim (\mu^e(P, c, u, t))^P}\left[UCQ(M^E; \mathbf{w})\right]$ for engagement-based optimization is strictly decreasing in $\gamma$.*

**Proof sketch of Theorem 3.** For sufficiently large values of $w_{costly}$, creators compete their utility down to 0, so the only remaining strategic choice is how they choose to trade off effort spent on gaming versus investment. If gaming is costly, then creators need to expend more of their effort on gaming to achieve a desired increase in engagement, so they will necessarily devote less effort to investment in quality. In contrast, if gaming is costless, creators devote all of their effort to investment. To formalize this intuition, we explicitly compute user consumption of quality using the equilibrium characterization. We defer the proof to Appendix E.1. □

Theorem 3 thus has a striking consequence for platform design: to improve user consumption of quality, it can help to reduce the costs of gaming tricks as much as possible. One concrete approach for reducing gaming costs is to increase the *transparency* of the platform's metric, for example by publishing the metric in an interpretable manner. In particular, if a content creator does not have access to the platform's metric, they would have to expend effort to learn the metric to game it; on the other hand, transparency would reduce these costs. Perhaps countuitively, our results suggest that increasing transparency can *improve* user consumption of quality in the presence of strategic content creators.[9] In particular, our results suggest the recent trend of recommender systems publishing their algorithms (e.g., Twitter [42]) may improve user consumption of quality content, and encourage the continued release of recommendation algorithms more broadly.

To further understand the impact of gaming costs $\gamma$, we compare the performance of engagement-based optimization with the performance of investment-based optimization (which does not depend on $\gamma$). We treat the performance of investment-based optimization as an "idealized baseline" for UCQ: the reason is that for any *fixed* content landscape $\mathbf{w}$, investment-based optimization maximizes the UCQ$(M; \mathbf{w})$ across all possible metrics $M$, because the objectives exactly align. The following result shows that engagement-based optimization performs strictly worse than investment-based optimization unless gaming tricks are *costless* (Figure 1b).

**Theorem 4.** *For any sufficiently large baseline utility $\alpha > -1$ and for bounded gaming costs $\gamma \in [0, 1)$, it holds that:*

$$\mathbb{E}_{\mathbf{w} \sim (\mu^e(P, c, u, t))^P}\left[UCQ(M^E; \mathbf{w})\right] \leq \mathbb{E}_{\mathbf{w} \sim (\mu^i(P, c, u, t))^P}\left[UCQ(M^I; \mathbf{w})\right],$$

*with equality if and only if $\gamma = 0$.*

Theorem 4 illustrates that reducing the gaming costs to 0 is *necessary* for engagement-based optimization to perform as well as the idealized baseline. This serves as a further motivation for a social planner to try to reduce gaming costs as much as possible, for example through increased transparency as discussed above.

## 4.2 User welfare

We next consider user utility realized by user consumption patterns, which can be interpreted as *user welfare*. We show that engagement-based optimization can alarmingly perform worse than random recommendations in terms of user welfare.[10]

We formalize user welfare by

$$\text{UW}(M; \mathbf{w}) := \mathbb{E}[u(w_{i^*(M;\mathbf{w})}, t) \cdot \mathbb{1}\left[u(w_{i^*(M;\mathbf{w})}, t) \geq 0\right]].$$

Taking into account the endogeneity of the content landscape, the user welfare at a symmetric mixed Nash equilibrium $\mu^M$ is $\mathbb{E}_{\mathbf{w} \sim (\mu^M)^P}\left[\text{UW}(M; \mathbf{w})\right]$.

The following result shows that engagement-based optimization always performs at least as poorly as random recommendations,

---

[9]This finding bears some resemblance to results in the strategic classification literature [5, 13]. For example, Ghalme et al. [13] shows that transparency is the optimal policy in terms of optimizing the decision-maker's accuracy. However, a lack of transparency is suboptimal in [13] because it prevents the decision-maker from being able to fully anticipating strategic behavior; in contrast, a lack of transparency is suboptimal in our setting because it leads effort to be spent on figuring how to game the classifier rather than investing in quality.

[10]We view random recommendations as a conservative baseline, since $M^R$ does not reward investment or gaming.

and can even perform strictly worse than random recommendations under certain conditions (Figure 1c).

**Theorem 5.** *Suppose that gaming costs $\gamma \in [0, 1)$ are bounded. If baseline utility $\alpha > 0$ is positive, the user welfare of engagement-based optimization is strictly lower than the user welfare of random recommendations:*

$$\mathbb{E}_{\mathbf{w} \sim (\mu^e(P,c,u,t))^P} [UW(M^E; \mathbf{w})] < \mathbb{E}_{\mathbf{w} \sim (\mu^r(P,c,u,t))^P} [UW(M^R; \mathbf{w})].$$

*If baseline utility $\alpha \leq 0$ is nonpositive, engagement-based optimization and random recommendations both result in zero user welfare:*

$$\mathbb{E}_{\mathbf{w} \sim (\mu^e(P,c,u,t))^P} [UW(M^E; \mathbf{w})] = \mathbb{E}_{\mathbf{w} \sim (\mu^r(P,c,u,t))^P} [UW(M^R; \mathbf{w})] = 0.$$

**Proof sketch of Theorem 5.** We first focus on the simple case where gaming tricks are free ($\gamma = 0$) and the baseline utility is positive ($\alpha \geq 0$). For engagement-based optimization, creators will increase gaming tricks until the user utility drops down to 0, which means the user welfare at equilibrium is 0. In contrast, for random recommendations, creators do not expend effort on either gaming tricks or investment; thus, the user welfare at equilibrium is $\alpha > 0$, which is strictly higher than the user welfare for engagement-based optimization. The other cases, though a bit more involved, follow from similar intuition: for engagement-based optimization, creators choose the balance between gaming tricks and investment in quality that drives user utility as close to zero as possible, whereas for random recommendations, creators choose the minimum amount of investment to achieve nonzero user utility. We defer the full proof to Appendix F. □

From a platform design perspective, Theorem 5 highlights the pitfalls of engagement-based optimization for users. In particular, the user welfare of engagement-based optimization can fall below the conservative baseline where users randomly select content on their own (and the content landscape shifts in response). This suggests that engagement-based optimization may not retain users in the long-run, especially in a competitive marketplace with multiple platforms.

## 4.3 Closed-form equilibrium characterizations for baseline approaches

Our analysis in Section 4.1 and Section 4.2 relied on the following closed-form equilibrium characterizations for investment-based optimization and random recommendations. To state these characterizations, we define a distribution $\mu^i(P, c, u, t)$ for investment-based optimization and a distribution $\mu^r(P, c, u, t)$ for random recommendations.

Since neither baseline approach directly incentivizes gaming tricks, the distributions $\mu^i(P, c, u, t)$ and $\mu^r(P, c, u, t)$ both satisfy $w_{\text{cheap}} = 0$ for all $w$ in the support (i.e., the marginal distribution of $W_{\text{cheap}}$ is a point mass at 0). We can thus convert the two-dimensional action space into a one-dimensional action space specified by $w_{\text{costly}}$, where the cost function is

$$C_b^I(w_{\text{costly}}) := c([w_{\text{costly}}, 0]) \tag{5}$$

and the utility function is:

$$U_b^I(w_{\text{costly}}, t) := u([w_{\text{costly}}, 0], t). \tag{6}$$

We now specify the marginal distribution of quality $W_{\text{costly}}$ for each baseline approach.

*Investment-based optimization.* We define the marginal distribution of $W_{\text{costly}}$ for $\mu^i(P, c, u, t)$ by:

$$\mathbb{P}[W_{\text{costly}} \leq w_{\text{costly}}] =$$
$$\min\left(1, \left(C_b^I(w_{\text{costly}})\right)^{1/(P-1)} \cdot \mathbb{1}\left[U_b^I(w_{\text{costly}}, t) \geq 0\right]\right).$$

We show that $\mu^i(P, c, u, t)$ is a symmetric mixed equilibrium.

**Theorem 6.** *The distribution $\mu^i(P, c, u, t)$ is a symmetric mixed Nash equilibrium in the game with $M = M^I$.*

*Random recommendations.* Let

$$\kappa := \min\left(\min_{w_{\text{costly}}} \left\{C_b^I(w_{\text{costly}}) \mid U_b^I(w_{\text{costly}}, t) \geq 0\right\}, 1\right)$$

be minimum cost to achieve 0 user utility, truncated at 1. Let the probability $\nu$ be defined as follows: $\nu = 0$ if $\kappa \leq 1/P$, and otherwise $\nu \in [0, 1]$ is the unique value such that that $\sum_{i=0}^{P-1} \nu^i = P \cdot \kappa$. We define the marginal distribution of $W_{\text{costly}}$ for $\mu^r(P, c, u, t)$ by

$$\mathbb{P}[W_{\text{costly}} = w_{\text{costly}}] =$$
$$\begin{cases} \nu & \text{if } w_{\text{costly}} = 0 \\ 1 - \nu & \text{if } w_{\text{costly}} = \operatorname{argmin}_{w'_{\text{costly}}} \left\{C_b^I(w'_{\text{costly}}) \mid U_b^I(w'_{\text{costly}}, t) \geq 0\right\} \\ 0 & \text{otherwise.} \end{cases}$$

We show that $\mu^r(P, c, u, t)$ is a symmetric mixed equilibrium.

**Theorem 7.** *The distribution $\mu^r(P, c, u, t)$ is a symmetric mixed Nash equilibrium in the game with $M = M^R$.*

We defer the proofs of Theorem 6 and Theorem 7 to Section 4.3.

## 5 Discussion

In this work, we study content creator competition for engagement-based recommendations that reward both quality and gaming tricks (e.g. clickbait). Our model further captures that a user only tolerates gaming tricks in sufficiently high-quality content, which also shapes content creator incentives. Our first result (Theorem 2) suggests that gaming and quality are complements for the content creators, which we empirically validate on a Twitter dataset. We then analyze the downstream performance of engagement-based optimization at equilibrium. We show that higher gaming costs can lead to lower average consumption of quality (Theorem 3) and the user welfare of engagement-based optimization can fall below that of random recommendations (Theorem 5).

More broadly, our results illustrate how content creator incentives can influence the downstream impact of a content recommender systems, which poses challenges when evaluating a platform's metric. In particular, there is a disconnect between how a platform's engagement metric behaves on a fixed content landscape and how the same metric behaves on an endogenous content landscape shaped by the metric. Interestingly, this disconnect manifests in two different performance measures relevant to society as a whole. We hope that our work encourages future evaluations of recommendation policies—for both of platform metrics and societal impacts—to carefully account for content creator incentives.

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

# A  Auxiliary definitions and lemmas

In our analysis of equilibria, it will be helpful to work with several quantities. We first define $C_t$ to be the set of content that achieves 0 utility for that type. That is:

$$C_t := \left\{ [w_{\text{costly}}, w_{\text{cheap}}] \mid u([w_{\text{costly}}, w_{\text{cheap}}], t) = 0. \right\}. \tag{7}$$

We also define an augmented version of these sets that also includes content with $w_{\text{costly}} = 0$ that achieving positive utility.

$$C_t^{\text{aug}} := \left\{ [w_{\text{costly}}, w_{\text{cheap}}] \mid u([w_{\text{costly}}, w_{\text{cheap}}], t) = 0 \right\} \cup \left\{ (0, w_{\text{cheap}}) \mid w_{\text{cheap}} \in [0, \min_{w' \mid u(w', t) = 0} w'_{\text{cheap}}] \right\}, \tag{8}$$

The set $C_t^{\text{aug}}$ turns out to be closely related to the function $f_t$ defined in (2).

**Lemma 8.** *The set $C_t^{aug}$ can be written as:*

$$C_t^{\text{aug}} = \left\{ [f_t(w_{\text{cheap}}), w_{\text{cheap}}] \mid w_{\text{cheap}} \geq 0 \right\}$$

*where $f_t$ is defined by (2).*

PROOF. First, we show that $C_t^{\text{aug}} \subseteq \left\{ (f_t(w_{\text{cheap}}), w_{\text{cheap}}) \mid w_{\text{cheap}} \geq 0 \right\}$. If $w \in C_t^{\text{aug}}$, then either $u(w, t) = 0$ or $w_{\text{costly}} = 0$ and $w_{\text{cheap}} \in [0, \min_{w' \mid u(w', t) = 0} w'_{\text{cheap}}]$. If $u(w, t) = 0$, since investing in quality is costly, it must hold that $w_{\text{costly}} = f_t(w_{\text{cheap}})$. Next, suppose that $w_{\text{cheap}} \in [0, \min_{w' \mid u(w', t) = 0} w'_{\text{cheap}}]$. We observe that $\min_{w' \mid u(w', t) = 0} w'_{\text{cheap}}$ is the unique value of $w'_{\text{cheap}}$ such that $u([0, w'_{\text{cheap}}], t) = 0$. This implies that $u([0, w_{\text{cheap}}], t) \geq 0$, so $f_t(w_{\text{cheap}}) = 0$ as desired.

Next, we show that $\left\{ [f_t(w_{\text{cheap}}), w_{\text{cheap}}] \mid w_{\text{cheap}} \geq 0 \right\} \subseteq C_t^{\text{aug}}$. Let $w = [f_t(w_{\text{cheap}}), w_{\text{cheap}}]$ for some $w_{\text{cheap}} \geq 0$. If $u(w, t) = 0$, then $w \in C_t^{\text{aug}}$ as desired. If $u(w, t) > 0$, then it must hold that $w_{\text{costly}} = 0$ (otherwise, it would be possible to lower $w_{\text{costly}}$ while keeping utility nonnegative, which contradicts the fact that $f_t(w_{\text{cheap}}) = w_{\text{costly}}$), so $w \in C_t^{\text{aug}}$. □

We prove that the function $f_t$ is weakly increasing.

**Lemma 9.** *The function $f_t$ as defined in (2) is weakly increasing. Moreover, the function $M^E([f_t(w_{cheap}), w_{cheap}])$ is strictly increasing in $w_{cheap}$.*

PROOF. Suppose that $w^1_{\text{cheap}} \geq w^2_{\text{cheap}}$. We claim that $f_t(w^1_{\text{cheap}}) \geq f_t(w^2_{\text{cheap}})$. To see this, note that

$$u([f_t(w^1_{\text{cheap}}), w^2_{\text{cheap}}]) > u([f_t(w^1_{\text{cheap}}), w^1_{\text{cheap}}]) \geq 0,$$

which proves the first statement.

To see that $M^E([f_t(w_{\text{cheap}}), w_{\text{cheap}}])$ is increasing, note that $f_t$ is a weakly increasing function (see Lemma 9) and that $M$ is strictly increasing in both of its arguments. □

Finally, we define an induced cost function for engagement, given by the optimization program

$$C_t^E(m) := \inf_{w \in \mathbb{R}^2_{\geq 0}} c(w) \text{ s.t. } u(w, t) \geq 0, M^{\text{platform}}(w) \geq m, \tag{9}$$

which captures the minimum production cost to create content with engagement at least $m$ and nonnegative user utility. We show the following properties of the optima of (9).

**Lemma 10.** *The optimization program $\inf_{w \in \mathbb{R}^2_{\geq 0}} c(w)$ s.t. $u(w, t) \geq 0, M^{platform}(w) \geq m$ satisfies the following properties:*

*(1) For any $m \in \left\{ M^E(w) \mid w \in C_t^{aug} \right\}$, the optimization program is feasible and any optimum $w^*$ satisfies $w^* \in C_t^{aug}$.*

*(2) If $m \in \left\{ M^E(w) \mid w \in C_t^{aug} \right\}$ and $C_t^E(m) > 0$, the optimization program has a unique optimum $w^*$ and moreover $M^{platform}(w^*) = m$.*

PROOF. Suppose that $m \in \left\{ M^E(w) \mid w \in C_t^{\text{aug}} \right\}$.

First, we show that the optimization program is feasible. Suppose that $w$ such that $M^E(w) = m$. Using the fact that $u([w'_{\text{costly}}, w_{\text{cheap}}], t)$ approaches $\infty$ as $w'_{\text{costly}} \to \infty$, we see that there exists $w'_{\text{costly}} \geq w_{\text{costly}}$ such that $M^E([w'_{\text{costly}}, w_{\text{cheap}}]) \geq M^E(w) = m$ and $u([w'_{\text{costly}}, w_{\text{cheap}}], t) \geq 0$, as desired.

Next, we show that there exists $w \in \mathbb{R}^2_{\geq 0}$ such that $u(w, t) \geq 0, M^E(w) \geq m$, and $c(w) = C_t^E(m)$. To make the domain compact, observe that there exists $w' \in \mathbb{R}^2_{\geq 0}$ such that $M^E(w') = m$ by assumption, which means that $C_t^E(m) \leq c(w')$. The set

$$\left\{ w \in \mathbb{R}^2_{\geq 0} \mid c(w) \leq c(w'), u(w, t) \geq 0, M^E(w) \geq m \right\} = \left\{ w \in \mathbb{R}^2_{\geq 0} \mid M^E(w) \geq m \right\} \cap \left\{ w \in \mathbb{R}^2_{\geq 0} \mid u(w, t) \geq 0 \right\} \cap c^{-1}\left([0, c(w')]\right).$$

The first two terms are closed, and the last term is compact (because the preimage of a continuous function of a compact set is compact). This means that the intersection is compact. Now, we use the fact that the inf of a continuous function over compact set is achievable.

Let $w^*$ be an optima. We show the following two properties:

(P1) If $w^*_{\text{costly}}, w^*_{\text{cheap}} > 0$, then $M^E(w^*) = m$.

(P2) If $w^*_{\text{costly}} > 0$, then $u(w^*, t) = 0$.

First, we show (P1). Assume for sake of contradiction that $M^E(w) > m$. Let $d$ be the direction normal to $\nabla u(w)$ where the costly coordinate is negative and the cheap coordinate is negative. We see that

$$\langle d, \nabla M^E(w^*)\rangle = -|d_1|(\nabla M^E(w^*))_1 - |d_2|(\nabla M^E(w^*))_2 < 0$$

$$\langle d, \nabla c(w^*)\rangle = -|d_1|(\nabla c(w^*))_1 - |d_2|(\nabla c(w^*))_2 < 0$$

$$\langle d, \nabla u(w^*)\rangle = 0.$$

This proves there exists $\epsilon > 0$ such that $w' = w + \epsilon d$ satisfies $M^E(w') \geq m$, $u(w', t) \geq 0$, and $c(w') < c(w^*)$, which is a contradiction.

Next, we show (P2). Assume for sake of contradiction that $u(w^*, t) > 0$. Let $d$ be the normal direction to $\nabla M^E(w^*)$ where the costly coordinate is negative and the cheap coordinate is positive. We see that

$$\langle d, \nabla u(w^*, t)\rangle = -|d_1|(\nabla u(w))_1 + d_2(\nabla u(w))_2 < 0$$

$$\langle d, \nabla M^E(w^*)\rangle = 0,$$

Moreover, we can see that $\langle d, \nabla c(w^*)\rangle = -|d_1|(\nabla c(w))_1 + d_2(\nabla c(w))_2 < 0$, since this can be written as:

$$\frac{(\nabla c(w))_1}{(\nabla c(w))_2} > \frac{|d_2|}{|d_1|} = \frac{(\nabla M^E(w))_1}{(\nabla M^E(w))_2},$$

which holds by assumption. This proves there exists $\epsilon > 0$ such that $w' = w + \epsilon d$ satisfies $M^E(w') \geq m$, $u(w', t) \geq 0$, and $c(w') < c(w^*)$, which is a contradiction.

We now show that $w^* \in C_t^{\text{aug}}$. First, suppose that $w^*_{\text{costly}} = 0$. Then, using the fact that $u(w^*, t) \geq 0$, we see that $f_t(w^*_{\text{cheap}}) = 0 = w_{\text{costly}}*$, so by Lemma 8, $w^* \in C_t^{\text{aug}}$. Next, suppose that $w^*_{\text{costly}} > 0$. Then we see that $u(w^*, t) = 0$ by (P2), so $w^* \in C_t^{\text{aug}}$.

For the remainder of the analysis, we assume that $c(w^*) = C_t^E(m) > 0$.

If gaming is costless $((\nabla(c(w)))_2 = 0$ for all $w)$ and $c(w^*) > 0$, then it must hold that $w^*_{\text{costly}} > 0$. This implies that $u(w^*, t) = 0$. This means that there is a unique value $w \in C_t^{\text{aug}}$ such that $c(w) = C_t^E(m)$, so this implies that $w^*$ is the unique optima. If $w^*_{\text{cheap}} > 0$, then we can apply (P1) to see that $M^E(w^*) = m$. If $w^*_{\text{cheap}} = 0$, the fact that $[0, w^*_{\text{costly}}] \in C_t^{\text{aug}}$ implies that $M^E(w^*) = \inf_{w \in C_t^{\text{aug}}} M^E(w)$. By the assumption that $m \in \{M^E(w) \mid w \in C_t^{\text{aug}}\}$, this means that $m = M^E(w^*)$ as desired.

If gaming is costly $((\nabla(c(w)))_2 = 0$ for all $w)$ and $C_t^E(m) > 0$, then there is a unique value $w \in C_t^{\text{aug}}$ such that $c(w) = c(w^*)$, which shows there is a unique optima. If $w^*_{\text{cheap}} > 0$ and $w^*_{\text{costly}} > 0$, then (P1) implies that $m = M^E(w^*)$. If $w^*_{\text{cheap}} = 0$, then the fact that $[0, w^*_{\text{costly}}] \in C_t^{\text{aug}}$ implies that $M^E(w^*) = \inf_{w \in C_t^{\text{aug}}} M^E(w)$. Finally, suppose that $w^*_{\text{costly}} = 0$. Assume for sake of contradiction that $M^E([0, w^*_{\text{cheap}}]) > m$. Then there exists $w_{\text{cheap}} < w^*_{\text{cheap}}$ such that $M^E([0, w_{\text{cheap}}]) \geq m$, $c([0, w'_{\text{cheap}}]) < c([0, w^*_{\text{cheap}}])$, and $u([0, w'_{\text{cheap}}], t) \geq u([0, w^*_{\text{cheap}}], t)$, which would mean that $w^*$ is not an optima, which is a contradiction. □

# B  Proofs for Section 2

Before proving Theorem 1, we prove the following properties of $\mu^e(P, c, u, t)$.

**Lemma 11.** *The distribution $\mu^e(P, c, u, t)$ satisfies the following properties:*

(P1) *The only possible atom in the distribution $\mu^e(P, c, u, t)$ is at $(0, 0)$, and moreover that $(0, 0)$ is an atom when $f_t(0) > 0$.*

(P2) *Suppose that $(w_{\text{cheap}}, w_{\text{costly}}) \in \text{supp}(\mu^e(P, c, u, t))$. If $(w_{\text{cheap}}, w_{\text{costly}}) \neq (0, 0)$ or if $f_t(0) = 0$, then it holds that $u([w_{\text{cheap}}, w_{\text{costly}}], t) \geq 0$.*

(P3) *If $(0, 0)$ is an atom of $\mu^e(P, c, u, t)$, then $u([0, 0], t) < 0$.*

PROOF. To prove (P1), note that if $(w_{\text{cheap}}, w_{\text{costly}}) \in \text{supp}(\mu^e(P, c, u, t))$ is an atom, then $w_{\text{cheap}}$ must be an atom in the marginal distribution $W_{\text{cheap}}$. The specification of the cdf shows that the only possible atom is at $W_{\text{cheap}} = 0$. Moreover, 0 is an atom of $W_{\text{cheap}}$ if and only if $c(f_t(0), 0) > 0$, which occurs if and only if $f_t(0) > 0$. When $f_t(0) > 0$, we further see that the conditional distribution $W_{\text{costly}}$ is a point mass at 0, as desired.

To prove (P2), note that $W_{\text{costly}}$ is a point mass at $f_t(w_{\text{cheap}})$. By the definition of $f_t$, it holds that $u([w_{\text{cheap}}, w_{\text{costly}}], t) \geq 0$.

To prove (P3), note that the first property showed that $(0, 0)$ is an atom if and only if $f_t(0) > 0$. By the definition of $f_t$, we see that $u([0, 0], t) < 0$ as desired.

□

We prove Theorem 1.

PROOF OF THEOREM 1. Let $\mu = \mu^e(P, c, u, t)$ for notational convenience. We analyze the expected utility of $H(w) = \mathbb{E}_{\mathbf{w}_{-i} \sim \mu_{-i}}[U_i(w; w_{-i})]$ of a content creator if all of the creators choose the strategy $\mu$. It suffices to show that any $w^* \in \text{supp}(\mu)$ is a best response $w^* \in \text{argmax}_w H(w)$. We use the properties (P1)-(P4) in Lemma 11.

First, we observe that we can write $H(w)$ as:

$$H(w) = \mathbb{E}_{\mathbf{w}_{-i} \sim \mu_{-i}}[U_i(w; w_{-i})]$$

$$= \mathbb{1}[u_t(w) \geq 0] \cdot \mathbb{P}_{W_{\text{cheap}}}[M^E(w) > M^E([f_t(W_{\text{cheap}}), W_{\text{cheap}}])]^{P-1} - c(w),$$

because (P1) implies that the only possible atom occurs at $[0,0]$, (P3) implies that $u([0,0], t) < 0$ if $[0,0]$ is an atom, and (P2) implies that $\mathbb{1}[u(w, t) \geq 0] = 0$ for $(w_{\text{cheap}}, w_{\text{costly}}) \neq (0,0)$.

If $w \in \text{supp}(\mu)$, then we claim that $H(w) = 0$. If $w_{\text{cheap}} = 0$ and $f_t(w_{\text{cheap}}) = 0$, it is immediate that $H(w) = 0$. Otherwise, if $w = [f_t(w_{\text{cheap}}), w_{\text{cheap}}]$. By (P2), it holds that $u_t(w) \geq 0$. This means that:

$$H(w) = \mathbb{P}_{W_{\text{cheap}}}[M^E([f_t(w_{\text{cheap}}), w_{\text{cheap}}]) > M^E([f_t(W_{\text{cheap}}), W_{\text{cheap}}])]^{P-1} - c(w)$$

$$=_{(1)} \mathbb{P}_{W_{\text{cheap}}}[w_{\text{cheap}} > W_{\text{cheap}}]^{P-1} - c(w)$$

$$= 0,$$

where (1) uses the fact that $M^E([f_t(w_{\text{cheap}}), w_{\text{cheap}}])$ is strictly increasing in $w_{\text{cheap}}$ (Lemma 9).

The remainder of the proof boils down to showing that $H(w) \leq 0$ for any $w$. If $u_t(w, t) < 0$, then $H(w) \leq 0$. If $u_t(w, t) \geq 0$, then

$$H(w) = \mathbb{P}_{W_{\text{cheap}}}[M^E(w) > M^E([f_t(W_{\text{cheap}}), W_{\text{cheap}}])]^{P-1} - c(w).$$

It suffices to show that $H(w) \leq 0$ at any best-response $w$ such that $u(w, t) \geq 0$. If $w$ is a best response and $u(w, t) \geq 0$, then it must be true that $w$ is a solution to (9). By Lemma 10, this means that $w \in C_t^{\text{aug}}$, and by Lemma 8, this means that $w$ is of the form $[f_t(w_{\text{cheap}}), w_{\text{cheap}}]$, which means that:

$$H(w) = \mathbb{P}_{W_{\text{cheap}}}[w_{\text{cheap}} > W_{\text{cheap}}]^{P-1} - c(w) \leq 0,$$

which proves the desired statement.                                                                                                      □

## C  Proofs for Section 3

To prove Theorem 2, we first show that the support of the equilibrium is contained in $C_t^{\text{aug}}$.

**Lemma 12.** *Let $(\mu_1, \ldots, \mu_P)$ be any mixed Nash equilibrium. Then $supp(\mu_i) \subseteq C_t^{aug} \cup \{w \mid c(w) = 0\}$ for all $i \in [P]$.*

PROOF. Assume for sake of contradiction that $w' \in \text{supp}(\mu_i)$ satisfies $w' \notin \cup_{t \in t} C_t^{\text{aug}} \cup \{w \mid c(w) = 0\}$. (This immediately implies that $c(w') > 0$.) We will show that $w'$ is not a best response. We split into two cases: (1) $u(w', t) < 0$, and (2) $u(w', t) \geq 0$.

First, suppose that $u(w', t) < 0$. Then the user will not consume the content. Since $c(w') > 0$, it holds that

$$\mathbb{E}_{\mathbf{w}_{-i} \sim \mu_{-i}}[U_i(w'; \mathbf{w}_{-i})] < 0.$$

However, if the creator were to instead choose the content $[0,0]$ which incurs $c([0,0]) = 0$ cost, they would get nonnegative utility:

$$\mathbb{E}_{\mathbf{w}_{-i} \sim \mu_{-i}}[U_i([0,0]; \mathbf{w}_{-i})] \geq 0 > \mathbb{E}_{\mathbf{w}_{-i} \sim \mu_{-i}}[U_i(w'; \mathbf{w}_{-i})].$$

Thus, $w'$ is not a best response, which is a contradiction.

Next, suppose that $u(w', t) \geq 0$. Let $m = M^E(w')$ be the engagement metric evaluated at $w'$. We see that the optimization program

$$\min_w c(w) \text{ s.t. } u(w, t) \geq 0, M^{\text{platform}}(w) \geq m$$

is feasible, since $w'$ satisfies the constraints by definition. Let $w^*$ be an optimal solution to the optimization program.

We first claim that $c(w^*) < c(w')$. To see this, first observe that $C_t^E(m) = c(w^*)$ by definition. If $C_t^E(m) = 0$, then:

$$c(w^*) = C_t^E(m) = 0 < c(w').$$

If $C_t^E(m) > 0$, then we apply Lemma 10 to see that $w^*$ is the unique optima, so $c(w^*) < c(w')$.

This means that:

$$\mathbb{E}_{\mathbf{w}_{-i} \sim \mu_{-i}}[U_i(w^*; \mathbf{w}_{-i})] > \mathbb{E}_{\mathbf{w}_{-i} \sim \mu_{-i}}[U_i(w'; \mathbf{w}_{-i})],$$

so $w'$ is not a best response which is a contradiction.                                                                                □

Since $w'$ is a best response, $w'$ must be a solution to the optimization program

$$\min_w c(w) \text{ s.t. } u(w, t) \geq 0, M^{\text{platform}}(w) \geq m.$$

We now prove Theorem 2 using Lemma 12 together with the properties in Appendix A.

PROOF OF THEOREM 2. By Lemma 12, the support of each $\mu_i$ satisfies:

$$\text{supp}(\mu_i) \subseteq C_t^{\text{aug}} \cup \{w \mid c(w) = 0\}.$$

Since gaming is costly by assumption, it holds that $\{w \mid c(w) = 0\} = \{[0, 0]\}$. We now apply Lemma 8 to see that:

$$\text{supp}(\mu_i) \subseteq \left\{[f_t(w_{\text{cheap}}), w_{\text{cheap}}]\right\} \cup \{(0, 0)\}.$$

The statement follows from the fact that $f_t$ is weakly increasing (Lemma 9). □

## D  Proofs for Section 4.3

We prove Theorem 6.

PROOF OF THEOREM 6. Let $\mu = \mu^{\text{i}}(P, c, u, t)$ for notational convenience, and let $(W_{\text{costly}}, W_{\text{cheap}}) \sim \mu$. We analyze the expected utility of

$$H(w) = \mathbb{E}_{\mathbf{w}_{-i} \sim \mu_{-i}}[U_i(w; w_{-i})]$$

of a content creator if all of the creators choose the strategy $\mu$. We show that $H(w) = 0$ if $w \in \text{supp}(\mu)$ and $H(w) \leq 0$ for any $w \in \mathbb{R}^2_{\geq 0}$.

Let $w \in \mathbb{R}^2_{\geq 0}$ be any content vector, and let $w' = [w_{\text{costly}}, 0]$ be the vector with identical quality but no gaming tricks. Since

$$U_b^I(w_{\text{costly}}, t) = u(w', t) \geq u(w, t),$$

$M^I(w) = M^I(w')$, and $c(w) \geq c(w')$, it holds that $H(w) \leq H(w')$. Since all $w'' \in \text{supp}(\mu)$ also satisfy $w''_{\text{cheap}} = 0$, we can restrict the rest of our analysis to $w$ such that $w_{\text{cheap}} = 0$.

Moreover, we see that:

$$
\begin{aligned}
H(w) &= \mathbb{E}_{\mathbf{w}_{-i} \sim \mu_{-i}}[U_i(w'; w_{-i})] \\
&= \left(\min\left(1, \left(C_b^I(w_{\text{costly}})\right) \cdot \mathbb{1}\left[U_b^I(w_{\text{costly}}, t) \geq 0\right]\right)\right) \cdot \mathbb{1}[U_b^I(w_{\text{costly}}, t)] - C_b^I(w_{\text{costly}}) \\
&\leq C_b^I(w_{\text{costly}}) - C_b^I(w_{\text{costly}}) \\
&= 0.
\end{aligned}
$$

Moreover, we see that if $w_{\text{costly}} \in \text{supp}(W_{\text{costly}})$, then either $w_{\text{costly}} = 0$ and $H(w) = 0$, or $w_{\text{costly}} > 0$ and

$$H(w) = \left(\min\left(1, \left(C_b^I(w_{\text{costly}})\right) \cdot \mathbb{1}\left[U_b^I(w_{\text{costly}}, t) \geq 0\right]\right)\right) \cdot \mathbb{1}[U_b^I(w_{\text{costly}}, t)] - C_b^I(w_{\text{costly}}) = \min\left(1, C_b^I(w_{\text{costly}})\right) - C_b^I(w_{\text{costly}}) = 0.$$

This proves that $\mu$ is an equilibrium as desired. □

We prove Theorem 7.

PROOF. Let $\mu = \mu^{\text{r}}(P, c, u, t)$ for notational convenience, and let $(W_{\text{costly}}, W_{\text{cheap}}) \sim \mu$. Moreover, let

$$w_{\text{costly}}^* = \underset{w'_{\text{costly}}}{\text{argmin}} \left\{C_b^I(w'_{\text{costly}}) \mid U_b^I(w'_{\text{costly}}, t) \geq 0\right\}.$$

We analyze the expected utility of

$$H(w) = \mathbb{E}_{\mathbf{w}_{-i} \sim \mu_{-i}}[U_i(w; w_{-i})]$$

of a content creator if all of the creators choose the strategy $\mu$. We show that $w \in \text{argmax}_{w'} H(w')$ for any $w \in \text{supp}(\mu)$.

Let $w \in \mathbb{R}^2_{\geq 0}$ be any content vector, and let $w' = [w_{\text{costly}}, 0]$ be the vector with identical quality but no gaming tricks. Since

$$U_b^I(w_{\text{costly}}, t) = u(w', t) \geq u(w, t),$$

$M^R(w) = M^R(w')$, and $c(w) \geq c(w')$, it holds that $H(w) \leq H(w')$. Since all $w'' \in \text{supp}(\mu)$ also satisfy $w''_{\text{cheap}} = 0$, we can restrict the rest of our analysis to $w$ such that $w_{\text{cheap}} = 0$.

We split into two cases: $\kappa \leq 1/P$ and $\kappa \in (1/P, 1]$.

If $\kappa \leq 1/P$, then $W_{\text{costly}}$ is a point mass at $w_{\text{costly}}^*$. Note that:

$$H(w) = \mathbb{E}_{\mathbf{w}_{-i} \sim \mu_{-i}}[U_i(w; w_{-i})] = \frac{\mathbb{1}\left[U_b^I(w_{\text{costly}}, t) \geq 0\right]}{P} - C_b^I(w) \leq \frac{1}{P} - \kappa.$$

Moreover, for $w_{\text{costly}} = w_{\text{costly}}^*$, it holds that $H(w) = \frac{1}{P} - \kappa$. This proves that $w \in \text{argmax}_{w'} H(w')$ for any $w \in \text{supp}(\mu)$, as desired.

If $\kappa \in (1/P, 1]$, then we see that $\nu$ is the unique value such $\sum_{i=1}^{P-1} \nu^i = P \cdot \kappa$. Note that $W_{\text{costly}}$ is $w_{\text{costly}}^*$ with probability $1 - \nu$ and 0 with probability $\nu$. Moreover, note that:

$$H(w) = \mathbb{E}_{\mathbf{w}_{-i} \sim \mu_{-i}}[U_i(w; w_{-i})] = \mathbb{1}\left[U_b^I(w_{\text{costly}}, t) \geq 0\right] \cdot \mathbb{E}_Y\left[\frac{1}{1+Y}\right] - C_b^I(w),$$

where $Y \sim \text{Bin}(P - 1, 1 - \nu)$ is distributed as a binomial random variable with success probability $1 - \nu$. (The second equality holds because $Y$ is distributed as the number of creators $j \neq i$ who choose content generating nonnegative utility for the user.) A simple calculation shows that:

$$\mathbb{E}_Y\left[\frac{1}{1 + Y}\right] = \frac{1}{P}\sum_{i=0}^{P-1} \nu^i = \kappa,$$

where the last equality follows from the definition of $\eta$. This means that $H(w) \leq 0$ for all $w$. For $w_{costly} = w^*_{costly}$ and $w_{costly} = 0$, it holds that $H(w) = 0$. This means that $w \in \text{argmax}_{w'} H(w')$ for any $w \in \text{supp}(\mu)$, as desired. □

# E Proofs for Section 4.1

## E.1 Proofs of Theorem 3 and Theorem 4

The main lemma is the following characterization of user consumption of utility as the maximum investment in quality across the content landscape.

**Lemma 13.** *Consider the setup of Theorem 3. For $\mathbf{w} \in supp(\mu^i(P, c, u, t))^P$, it holds that*

$$UCQ(M^I; \mathbf{w}) = \max_{w \in \mathbf{w}} w_{costly}$$

*and for $\mathbf{w} \in supp(\mu^e(P, c, u, t))^P$, it holds that*

$$UCQ(M^E; \mathbf{w}) = \max_{w \in \mathbf{w}} w_{costly}.$$

We now prove Lemma 13.

PROOF OF LEMMA 13. We observe that for $w \in \text{supp}(\mu^i(P, c, u, t))$, it holds that if $\mathbb{1}[u(w, t)] < 0$, then $w = [0, 0]$. Thus, for $\mathbf{w} \in \text{supp}(\mu^i(P, c, u, t))^P$, it holds that:

$$UCQ(M^I; \mathbf{w}) = \mathbb{E}\left[w^{costly}_{i^*(M^I;\mathbf{w})} \cdot \mathbb{1}[u(w_{i^*(M;\mathbf{w})}, t) \geq 0]\right] = \mathbb{E}\left[w^{costly}_{i^*(M^I;\mathbf{w})}\right].$$

Moreover, since $w_{cheap} = 0$ for all $w \in \text{supp}(\mu^i(P, c, u, t))$ and by the definition of $M^I$, we see that $w^{costly}_{i^*(M^I;\mathbf{w})} = \max_{w \in \mathbf{w}} w_{costly}$. This means that:

$$UCQ(M^I; \mathbf{w}) = \mathbb{E}\left[\max_{w \in \mathbf{w}} w_{costly}\right].$$

Similarly, we observe that for $w \in \text{supp}(\mu^e(P, c, u, t))$, it holds that if $\mathbb{1}[u(w, t)] < 0$, then $w = [0, 0]$. Thus, for $\mathbf{w} \in \text{supp}(\mu^e(P, c, u, t))^P$, it holds that:

$$UCQ(M^E; \mathbf{w}) = \mathbb{E}\left[w^{costly}_{i^*(M;\mathbf{w})} \cdot \mathbb{1}[u(w_{i^*(M;\mathbf{w})}, t)] \geq 0\right] = \mathbb{E}\left[w^{costly}_{i^*(M;\mathbf{w})}\right].$$

Moreover, by the definition of $\text{supp}(\mu^e(P, c, u, t))$ and by the definition of $M^E$, we see that $w^{costly}_{i^*(M^E;\mathbf{w})} = \max_{w \in \mathbf{w}} w_{costly}$. This means that:

$$UCQ(M^E; \mathbf{w}) = \mathbb{E}\left[\max_{w \in \mathbf{w}} w_{costly}\right].$$

□

Using Lemma 13, we prove Theorem 3 and Theorem 4.

PROOF OF THEOREM 3 AND THEOREM 4. By Lemma 13, it suffices to analyze

$$\mathbb{E}_{\mathbf{w}\sim\mu^P}\left[\max_{w \in \mathbf{w}} w_{costly}\right]$$

where $\mu \in \left\{\mu^e(P, c, u, t), \mu^i(P, c, u, t)\right\}$.

To analyze these expressions, let $\beta_t = \min\left\{w_{costly} \mid u([w_{costly}, 0]) \geq 0\right\}$. If $(W_{costly}, W_{cheap}) \sim \mu^e(P, c, u, t)$, then:

$$\mathbb{P}[W_{costly} \leq w_{costly}] = \begin{cases} (\min(1, c([\beta_t, 0])))^{1/(P-1)} & \text{if } 0 \leq w_{costly} \leq \beta_t \\ \left(\min(1, c([w_{costly}, f_t^{-1}(w_{costly})]))\right)^{1/(P-1)} & \text{if } w_{costly} \geq \beta_t. \end{cases}$$

If $(W_{costly}, W_{cheap}) \sim \mu^i(P, c, u, t)$, then:

$$\mathbb{P}[W_{costly} \leq w_{costly}] = \begin{cases} (\min(1, c([\beta_t, 0])))^{1/(P-1)} & \text{if } 0 \leq w_{costly} \leq \beta_t \\ \left(\min(1, c([w_{costly}, 0]))\right)^{1/(P-1)} & \text{if } w_{costly} \geq \beta_t. \end{cases}$$

Using the specification in Example 1, where $c([w_{\text{costly}}, 0]) = w_{\text{costly}}$ and $u(w, t) = w_{\text{costly}} - (w_{\text{cheap}}/t) + \alpha$, we can simplify these expressions. In particular, we see that $\beta_t = \max(0, -\alpha) \leq 1$ (since $\alpha > -1$ by assumption). This means that if $(W_{\text{costly}}, W_{\text{cheap}}) \sim \mu^e(P, c, u, t)$, then:

$$\mathbb{P}_{(W_{\text{costly}}, W_{\text{cheap}}) \sim (\mu^e(P,c,u,t))}[W_{\text{costly}} \leq w_{\text{costly}}] = \begin{cases} (-\alpha)^{1/(P-1)} & \text{if } 0 \leq w_{\text{costly}} \leq -\alpha \\ \left( \min(1, w_{\text{costly}} + \gamma \cdot f_t^{-1}(w_{\text{costly}})] ) \right)^{1/(P-1)} & \text{if } w_{\text{costly}} \geq \max(0, -\alpha). \end{cases}$$

Moreover, if $(W_{\text{costly}}, W_{\text{cheap}}) \sim \mu^i(P, c, u, t)$, then:

$$\mathbb{P}_{(W_{\text{costly}}, W_{\text{cheap}}) \sim (\mu^i(P,c,u,t))}[W_{\text{costly}} \leq w_{\text{costly}}] = \begin{cases} (-\alpha)^{1/(P-1)} & \text{if } 0 \leq w_{\text{costly}} \leq -\alpha \\ \left( \min(1, w_{\text{costly}}) \right)^{1/(P-1)} & \text{if } w_{\text{costly}} \geq \max(0, -\alpha). \end{cases}$$

*Proof of Theorem 3.* The marginal distribution of $W_{\text{costly}}$ for $\mu^e(P, c, u, t)$ implies for engagement-based optimization, the distribution of $W_{\text{costly}}$ for higher values of $\gamma$ is stochastically dominated by the distribution of $W_{\text{costly}}$ for lower values of $\gamma$. This implies that $\mathbb{E}_{\mathbf{w} \sim (\mu^e(P,c,u,t))^P}[\max_{w \in \mathbf{w}} w_{\text{costly}}]$ is strictly increasing in $\gamma$, which implies that $\mathbb{E}_{\mathbf{w} \sim (\mu^e(P,c,u,t))^P}[\text{UCQ}(M^E; \mathbf{w})]$ is strictly increasing in $\gamma$.

*Proof of Theorem 4.* Observe that the marginal distribution of $W_{\text{costly}}$ for $\mu^i(P, c, u, t)$ stochastically dominates the distribution of $W_{\text{costly}}$ for $\mu^e(P, c, u, t)$, with strict stochastic dominance. This implies that if $\gamma > 0$:

$$\mathbb{E}_{\mathbf{w} \sim (\mu^e(P,c,u,t))^P} \left[ \max_{w \in \mathbf{w}} w_{\text{costly}} \right] < \mathbb{E}_{\mathbf{w} \sim (\mu^i_{P,\alpha,t})^P} \left[ \max_{w \in \mathbf{w}} w_{\text{costly}} \right],$$

which implies that

$$\mathbb{E}_{\mathbf{w} \sim (\mu^e(P,c,u,t))^P} \left[ \text{UCQ}(M^E; \mathbf{w}) \right] < \mathbb{E}_{\mathbf{w} \sim (\mu^i_{P,\alpha,t})^P} \left[ \text{UCQ}(M^I; \mathbf{w}) \right].$$

Moreover, if $\gamma = 0$, the two distributions are equal, which implies that

$$\mathbb{E}_{\mathbf{w} \sim (\mu^e(P,c,u,t))^P} \left[ \max_{w \in \mathbf{w}} w_{\text{costly}} \right] = \mathbb{E}_{\mathbf{w} \sim (\mu^i_{P,\alpha,t})^P} \left[ \max_{w \in \mathbf{w}} w_{\text{costly}} \right],$$

which implies that

$$\mathbb{E}_{\mathbf{w} \sim (\mu^e(P,c,u,t))^P} \left[ \text{UCQ}(M^E; \mathbf{w}) \right] = \mathbb{E}_{\mathbf{w} \sim (\mu^i_{P,\alpha,t})^P} \left[ \text{UCQ}(M^I; \mathbf{w}) \right]. \qquad \square$$

# F  Proofs for Section 4.2

The proof of Theorem 5 follows from the following characterizations of the realized user utility for engagement-based optimization (Lemma 15) and random recommendations (Lemma 14) , stated and proved below.

**Lemma 14.** *Consider the setup of Theorem 5. Then it holds that:*

$$\mathbb{E}_{\mathbf{w} \sim (\mu^r(P,c,u,t))^P}[UW(M^R; \mathbf{w})] = \begin{cases} \alpha & \text{if } \alpha > 0 \\ 0 & \text{if } \alpha \leq 0. \end{cases}$$

PROOF. If $\alpha > 0$, then we see that $U_b^I(0, t) = \alpha$. This means that:

$$\min_{w_{\text{costly}}} \left\{ C_b^I(w_{\text{costly}}) \mid U_b^I(w_{\text{costly}}, t) \geq 0 \right\} = 0$$

and moreover the min is achieved at $w = [0, 0]$. This means that $\nu = 0$ and $\mu^r(P, c, u, t)$ is a point mass at $[0, 0]$. This means that:

$$\mathbb{E}_{\mathbf{w} \sim (\mu^r(P,c,u,t))^P}[UW(M^R; \mathbf{w})] = U_b^I(0, t) = \alpha.$$

If $\alpha \leq 0$, then

$$w_{\text{costly}}^* := \underset{w'_{\text{costly}}}{\text{argmin}} \left\{ C_b^I(w'_{\text{costly}}) \mid U_b^I(w'_{\text{costly}}, t) \geq 0 \right\}$$

satisfies $U_b^I(w_{\text{costly}}^*, t) = 0$. This means that if $(W_{\text{costly}}, W_{\text{cheap}}) \sim \mu^r(P, c, u, t)$, it holds that $\text{supp}(W_{\text{costly}}) \subseteq \left\{ w_{\text{costly}}^*, 0 \right\}$. Moreover, for any content landscape $\mathbf{w} \in \text{supp}(\mu^r(P, c, u, t))^P$, we see that:

$$\text{UW}(M^R; \mathbf{w}) := \mathbb{E}[u(w_{i^*(M^R; \mathbf{w})}, t) \cdot \mathbb{1}[u(w_{i^*(M^R; \mathbf{w})}, t) \geq 0]] = 0.$$

This means that:

$$\mathbb{E}_{\mathbf{w} \sim (\mu^r(P,c,u,t))^P}[UW(M^R; \mathbf{w})] = U_b^I(w_{\text{costly}}^*) = 0. \qquad \square$$

**Lemma 15.** *Consider the setup of Theorem 5. If $\alpha > 0$, then it holds that:*

$$\mathbb{E}_{\mathbf{w} \sim (\mu^e(P,c,u,t))^P}[UW(M^E; \mathbf{w})] < \alpha.$$

*If $\alpha \leq 0$, then it holds that:*

$$\mathbb{E}_{\mathbf{w} \sim (\mu^e(P,c,u,t))^P}[UW(M^E; \mathbf{w})] = 0.$$

Proof. First, suppose that $\alpha \leq 0$. In this case, we see that $u(w, t) \leq 0$ for all $w \in \text{supp}(\mu^e(P, c, u, t))^P)$. This implies that for any content landscape $\mathbf{w} \in \text{supp}(\mu^e(P, c, u, t))^P$, we see that:

$$UW(M^E; \mathbf{w}) := \mathbb{E}[u(w_{i^*(M^E; \mathbf{w})}, t) \cdot \mathbb{1}[u(w_{i^*(M^E; \mathbf{w})}, t) \geq 0]] = 0.$$

This means that:

$$\mathbb{E}_{\mathbf{w} \sim (\mu^e(P,c,u,t))^P}[UW(M^E; \mathbf{w})] = 0.$$

For $\alpha > 0$, we see that $w = [0, 0]$ is the unique value such that $w \in C_t^{\text{aug}}$ and $u(w, t) \geq \alpha$. Moreover, by Lemma 8, we know that $\{[f_t(w_{\text{cheap}}), w_{\text{cheap}}] \mid w_{\text{cheap}} \geq 0\} = C_t^{\text{aug}}$. We observe that $\text{supp}(\mu^e(P, c, u, t))^P$ is contained in $\{[f_t(w_{\text{cheap}}), w_{\text{cheap}}] \mid w_{\text{cheap}} \geq 0\} = C_t^{\text{aug}}$. This means that $u(w, t) < \alpha$ for all $w \in \text{supp}(\mu^e(P, c, u, t))^P$ such that $w \neq [0, 0]$. Since there is no point mass at 0, this means that the probability $[0, 0]$ shows up in the content landscape is 0, so

$$\mathbb{P}[UW(M^E; \mathbf{w}) < \alpha] = \mathbb{P}\left[\mathbb{E}[u(w_{i^*(M^E; \mathbf{w})}, t) \cdot \mathbb{1}[u(w_{i^*(M^E; \mathbf{w})}, t) \geq 0]] < \alpha\right] = 1.$$

This means that:

$$\mathbb{E}_{\mathbf{w} \sim (\mu^e(P,c,u,t))^P}[UW(M^E; \mathbf{w})] < \alpha.$$

$\square$

Using Lemma 15 and Lemma 14, we prove Theorem 5.

Proof of Theorem 5. We apply Lemma 15 and Lemma 14. When $\alpha > 0$, we see that:

$$\mathbb{E}_{\mathbf{w} \sim (\mu^e(P,c,u,t))^P}[UW(\mu^e; \mathbf{w})] < \alpha = \mathbb{E}_{\mathbf{w} \sim (\mu^r(P,c,u,t))^P}[UW(M^R; \mathbf{w})].$$

When $\alpha \leq 0$, we see that:

$$\mathbb{E}_{\mathbf{w} \sim (\mu^e(P,c,u,t))^P}[UW(\mu^e; \mathbf{w})] = 0 = \mathbb{E}_{\mathbf{w} \sim (\mu^r(P,c,u,t))^P}[UW(M^R; \mathbf{w})]$$

$\square$

Received 20 February 2007; revised 12 March 2009; accepted 5 June 2009

