# OpenReview forum: "Clickbait vs. Quality: How Engagement-Based Optimization Shapes the Content Landscape in Online Platforms"
_ACM.org/TheWebConf/2024/Conference — TheWebConf24_

### Official Review · Reviewer_3iRD · 2023-11-23

**Novelty:** 5
**Technical Quality:** 5

**Review:**

In this paper, the authors studied how engagement-based optimization impacted the contents landscape by specifically focusing on two concepts---quality and gaming tricks. The paper shows that under the equilibrium setting, there exists a positive correlation between quality and gaming (i.e., higher quality contents usually come along with more gaming tricks). It then studies the performance of engagement-based optimization under equilibrium, in terms of both the quality of contents consumed by the users and the user welfare, which shed lights on the importance of considering content creator incentives when a platform determines its metric.

Strength:
- This paper is well-written and well motivated using real-world examples.
- The topic that the paper studies has practical relevance. The insights provided in this paper are also interesting in that it suggests the correlation between contents quality and gaming, which provides novel insights to a platform's decision-making. Overall I find Theorem 3, 4 and 5 interesting in that they suggest that the platform should not impose high gaming cost, but rather allow content creators some room to perform gaming tricks while making quality contents. These are good managerial insights to have.
- The theoretical results of this paper appear sound.

Weakness & questions:
- My main concern regarding the current model lies in its simplicity. As the authors also suggested, the current model focuses on making one recommendation to a single user at a time, which can be too simplistic in practice. How does the model capture "a stream of contents and a population of homogeneous users"?
- The authors made certain assumption about the cost of content creation function as well as the platform engagement functions. I wonder if the authors can provide some justifications for these assumptions and potentially discuss extensions of the current results when the metrics considered by the platform deviate from such assumptions.
- As the authors mainly focused on example 1 in their theoretical discussion & numerical evaluation, I wonder if the authors also have similar results for examples 2 & 3.

**Questions:**

See above.

**Reviewer Confidence:**

3: The reviewer is confident but not certain that the evaluation is correct

**Scope:**

3: The work is somewhat relevant to the Web and to the track, and is of narrow interest to a sub-community

---

### Official Review · Reviewer_mEqL · 2023-11-28

**Novelty:** 4
**Technical Quality:** 4

**Review:**

### Summary
This paper introduces a framework to analyze content creators' behaviors in online content platforms, focusing on engagement-based optimization. The authors explore a scenario where creators choose between enhancing quality and employing gaming tactics. They establish a symmetric mixed Nash equilibrium, demonstrating a positive correlation between quality and gaming, and assess the impact of engagement optimization on user consumption and welfare.

### Strengths
1. The paper addresses a significant issue in online content platforms, examining the motivations and incentives of content creators.
2. The theoretical results are sound and interesting.
3. The paper offers valuable insights, notably the observed positive correlation between content quality and gaming strategies.

### Weaknesses
1. The main findings rely on a simplified model with two content dimensions: gaming and quality. Can these results be generalized to scenarios with multiple dimensions related to gaming and quality?
2. The symmetric mixed Nash equilibrium characterized in Theorem 1 raises the question of its uniqueness. If it is not unique, how can all possible symmetric mixed Nash equilibria be characterized?
3. Considering the unrealistic nature of symmetric strategies in practical scenarios, is it possible to extend the analysis to asymmetric mixed Nash equilibria?
4. The assumption of homogeneous content creators, all sharing identical cost functions for quality and gaming, seems unrealistic. Could the analysis be adapted to accommodate heterogeneity in cost functions among different creators?
5. There is a minor typo in Line 287: "observe observe" should be corrected to "observe".

**Questions:**

See the review part.

**Reviewer Confidence:**

3: The reviewer is confident but not certain that the evaluation is correct

**Scope:**

4: The work is relevant to the Web and to the track, and is of broad interest to the community

---

### Official Review · Reviewer_gS9S · 2023-11-28

**Novelty:** 6
**Technical Quality:** 5

**Review:**

The work studies a game between content creators competing based on engagement metrics and analyzes the equilibrium decisions about investment in quality and gaming tricks (e.g., clickbait).

The work theoretically and empirically proves higher-quality content typically exhibits higher gaming tricks.
Two downstream findings:
1. average quality of content consumed by users and show that it can decrease as gaming tricks become more costly for creators
2. engagement-based optimization can lead to lower user welfare at equilibrium than the conservative baseline of randomly recommending content

Strength:
The problem formulation is well-explained and rooted in game theory.
The insights drawn from the mathematical model are clear and interesting.

I have read the authors' rebuttal to my and other reviewers' comments. I will leave my rating unchanged.

**Questions:**

Line 346: “Our model makes the simplifying assumption that user choice is noiseless.”
Does it mean that if users' actions are irrational or change over time or via a shared account, the model in this work does not stand?

Usually, the exposure of the items (in the user’s feed) is exponentially discounted for lower-ranked items. How does the model proposed in the paper account for this?

Is “producers” and “content creators” the same group of people in this paper?

Typos:

Line 282 “the”
Line 287 “observe”
Line 690 “the”
Line 877 the square and round brackets are not paired.

**Reviewer Confidence:**

2: The reviewer is willing to defend the evaluation, but it is likely that the reviewer did not understand parts of the paper

**Scope:**

4: The work is relevant to the Web and to the track, and is of broad interest to the community

---

### Decision · Program_Chairs · 2024-01-22

**Decision:**

Accept

**Comment:**

The strengths identified by the reviewers are as follows:

 - The paper studies an interesting and practically relevant problem

 - The insights provided by the paper, particularly the positive correlation between content quality and gaming of content creators, are clear, economically interesting, and non-obvious.

 - The modeling and theoretical analysis is sound.

 The weaknesses identified by the reviewers are as follows:

 - It is not clear how generalizable the insights are, due to simplicity of the model, the focus on symmetric Nash equilibria, and assumptions about costs.

 I agree with the strengths identified by the reviewers. Regarding the weaknesses, my view is that the simplicity of the model is a strength if the authors can provide some justification that their insights are general and don't rely on the simplifying assumptions. In addition, none of the reviewers mentioned the empirical validation of the main insight of the paper on the Twitter dataset, perhaps because this part of the paper was not particularly strong (e.g. one weakness of this section is the features in the data, such as 'feed', 'angriness level', and 'number of favorites', don't map particularly well to the model in the paper).

 Overall, given the broadly appealing and somewhat surprising insights and sound analysis, I weakly recommend this paper for acceptance. A revised version that provides stronger empirical evidence (can be anecdotal, but needs to be better aligned with model) and more detail on generalizability of insights beyond the simple model and symmetric equilibrium (can be numerical analysis) would more comfortably clear the bar for acceptance.